# MULTI-AGENT COOPERATION
# AND THE EMERGENCE OF (NATURAL) LANGUAGE

**Angeliki Lazaridou**[1]*, **Alexander Peysakhovich**[2], **Marco Baroni**[2,3]
[1]Google DeepMind, [2]Facebook AI Research, [3]University of Trento
`angeliki@google.com, {alexpeys,mbaroni}@fb.com`

## ABSTRACT

The current mainstream approach to train natural language systems is to expose them to large amounts of text. This passive learning is problematic if we are interested in developing *interactive* machines, such as conversational agents. We propose a framework for language learning that relies on multi-agent communication. We study this learning in the context of referential games. In these games, a sender and a receiver see a pair of images. The sender is told one of them is the target and is allowed to send a message from a fixed, arbitary vocabulary to the receiver. The receiver must rely on this message to identify the target. Thus, the agents develop their own language interactively out of the need to communicate. We show that two networks with simple configurations are able to learn to coordinate in the referential game. We further explore how to make changes to the game environment to cause the "word meanings" induced in the game to better reflect intuitive semantic properties of the images. In addition, we present a simple strategy for grounding the agents' code into natural language. Both of these are necessary steps towards developing machines that are able to communicate with humans productively.

## 1 INTRODUCTION

> I tried to break it to him gently [...] the only way to learn an unknown language is to interact with a native speaker [...] asking questions, holding a conversation, that sort of thing [...] If you want to learn the aliens' language, someone [...] will have to talk with an alien. Recordings alone aren't sufficient.
>
> Ted Chiang, *Story of Your Life*

One of the main aims of AI is to develop agents that can cooperate with others to achieve goals (Wooldridge, 2009). Such coordination requires communication. If the coordination partners are to include humans, the most obvious channel of communication is natural language. Thus, handling natural-language-based communication is a key step toward the development of AI that can thrive in a world populated by other agents.

Given the success of deep learning models in related domains such as image captioning or machine translation (e.g., Sutskever et al., 2014; Xu et al., 2015), it would seem reasonable to cast the problem of training conversational agents as an instance of supervised learning (Vinyals & Le, 2015). However, training on "canned" conversations does not allow learners to experience the interactive aspects of communication. Supervised approaches, which focus on the structure of language, are an excellent way to learn general statistical associations between sequences of symbols. However, they do not capture the functional aspects of communication, i.e., that humans use words to coordinate with others and make things happen (Austin, 1962; Clark, 1996; Wittgenstein, 1953).

This paper introduces the first steps of a research program based on *multi-agent coordination communication games*. These games place agents in simple environments where they need to develop a language to coordinate and earn payoffs. Importantly, the agents start as blank slates, but, by playing a game together, they can develop and bootstrap knowledge on top of each others, leading to the emergence of a language.

---

*Work done while at Facebook AI Research.

The central problem of our program, then, is the following: How do we design environments that foster the development of a language that is portable to new situations and to new communication partners (in particular humans)?

We start from the most basic challenge of using a language in order to *refer* to things in the context of a two-agent game. We focus on two questions. First, whether *tabula rasa* agents succeed in communication. Second, what features of the environment lead to the development of codes resembling human language.

We assess this latter question in two ways. First, we consider whether the agents associate general conceptual properties, such as broad object categories (as opposed to low-level visual properties), to the symbols they learn to use. Second, we examine whether the agents' "word usage" is partially interpretable by humans in an online experiment.

Other researchers have proposed communication-based environments for the development of coordination-capable AI. Work in multi-agent systems has focused on the design of pre-programmed communication systems to solve specific tasks (e.g., robot soccer, Stone & Veloso 1998). Most related to our work, Sukhbaatar et al. (2016) and Foerster et al. (2016) show that neural networks can evolve communication in the context of games without a pre-coded protocol. We pursue the same question, but further ask how we can change our environment to make the emergent language more interpretable.

Others (e.g., the SHRLDU program of Winograd 1971 or the game in Wang et al. 2016) propose building a communicating AI by putting humans in the loop from the very beginning. This approach has benefits but faces serious scalability issues, as active human intervention is required at each step. An attractive component of our game-based paradigm is that humans may be added as players, but do not need to be there all the time.

A third branch of research focuses on "Wizard-of-Oz" environments, where agents learn to play games by interacting with a complex scripted environment (Mikolov et al., 2015). This approach gives the designer tight control over the learning curriculum, but imposes a heavy engineering burden on developers. We also stress the importance of the environment (game setup), but we focus on simpler environments with multiple agents that force them to get smarter by bootstrapping on top of each other.

We leverage ideas from work in linguistics, cognitive science and game theory on the emergence of language (Wagner et al., 2003; Skyrms, 2010; Crawford & Sobel, 1982; Crawford, 1998). Our game is a variation of Lewis' signaling game (Lewis, 1969). There is a rich tradition of linguistic and cognitive studies using similar setups (e.g., Briscoe, 2002; Cangelosi & Parisi, 2002; Spike et al., 2016; Steels & Loetzsch, 2012). What distinguishes us from this literature is our aim to, eventually, develop practical AI. This motivates our focus on more realistic input data (a large collection of noisy natural images) and on trying to align the agents' language with human intuitions.

Lewis' classic games have been studied extensively in game theory under the name of "cheap talk". These games have been used as models to study the evolution of language both theoretically and experimentally (Crawford, 1998; Blume et al., 1998; Crawford & Sobel, 1982). A major question in game theory is whether equilibrium actually occurs in a game as convergence in learning is not guaranteed (Fudenberg & Peysakhovich, 2014; Roth & Erev, 1995). And, if an equilibrium is reached, which one it will be (since they are typically not unique). This is particularly true for cheap talk games, which exhibit Nash equilibria in which precise language emerges, others where vague language emerges and others where no language emerges at all (Crawford & Sobel, 1982). In addition, because in these games language has no ex-ante meaning and only emerges in the context of the equilibrium, some of the emergent languages may not be very natural. Our results speak to both the convergence question and the question of what features of the game cause the appearance of different types of languages. Thus, our results are also of interest to game theorists.

An evolutionary perspective has recently been advocated as a way to mitigate the data hunger of traditional supervised approaches (Goodfellow et al., 2014; Silver et al., 2016). This research confirms that learning can be bootstrapped from *competition* between agents. We focus, however, on *cooperation* between agents as a way to foster learning while reducing the need for annotated data.

## 2 GENERAL FRAMEWORK

Our general framework includes K players, each parametrized by $\theta_k$, a collection of tasks/games that the players have to perform, a communication protocol $V$ that enables the players to communicate with each other, and payoffs assigned to the players as a deterministic function of a well-defined goal. In this paper we focus on a particular version of this: *referential games*. These games are structured as follows.

1. There is a set of images represented by vectors $\{i_1, \ldots, i_N\}$, two images are drawn at random from this set, call them $(i_L, i_R)$, one of them is chosen to be the "target" $t \in \{L, R\}$

2. There are two players, a sender and a receiver, each seeing the images - the sender receives input $\theta_S(i_L, i_R, t)$

3. There is a *vocabulary* $V$ of size $K$ and the sender chooses one symbol to send to the receiver, we call this the sender's policy $s(\theta_S(i_L, i_R, t)) \in V$

4. The receiver does not know the target, but sees the sender's symbol and tries to guess the target image. We call this the receiver's policy $r(i_L, i_R, s(\theta_S(i_L, i_R, t))) \in \{L, R\}$

5. If $r(i_L, i_R, s(\theta_S(i_L, i_R, t)) = t$, that is, if the receiver guesses the target, both players receive a payoff of 1 (win), otherwise they receive a payoff of 0 (lose).

Many extensions to the basic referential game explored here are possible. There can be more images, or a more sophisticated communication protocol (e.g., communication of a sequence of symbols or multi-step communication requiring back-and-forth interaction[1]), rotation of the sender and receiver roles, having a human occasionally playing one of the roles, etc.

## 3 EXPERIMENTAL SETUP

**Images** We use the McRae et al.'s (2005) set of 463 base-level concrete concepts (e.g., *cat, apple, car...*) spanning across 20 general categories (e.g., *animal*, *fruit/vegetable*, *vehicle...*). We randomly sample 100 images of each concept from ImageNet (Deng et al., 2009). To create target/distractor pairs, we randomly sample two concepts, one image for each concept and whether the first or second image will serve as target. We apply to each image a forward-pass through the pre-trained VGG ConvNet (Simonyan & Zisserman, 2014), and represent it with the activations from either the top 1000-D softmax layer (*sm*) or the second-to-last 4096-D fully connected layer (*fc*).

**Agent Players** Both sender and receiver are simple feed-forward networks. For the sender, we experiment with the two architectures depicted in Figure 1. Both sender architectures take as input the target (marked with a green square in Figure 1) and distractor representations, always in this order, so that they are implicitly informed of which image is the target (the receiver, instead, sees the two images in random order).

The *agnostic* sender is a generic neural network that maps the original image vectors onto a "game-specific" embedding space (in the sense that the embedding is learned while playing the game) followed by a sigmoid nonlinearity. Fully-connected weights are applied to the embedding concatenation to produce scores over vocabulary symbols.

The *informed* sender also first embeds the images into a "game-specific" space. It then applies 1-D convolutions ("filters") on the image embeddings by treating them as different channels. The informed sender uses convolutions with kernel size 2x1 applied dimension-by-dimension to the two image embeddings (in Figure 1, there are 4 such filters). This is followed by the sigmoid nonlinearity. The resulting feature maps are combined through another filter (kernel size $f$x1, where $f$ is the number of filters on the image embeddings), to produce scores for the vocabulary symbols. Intuitively, the informed sender has an inductive bias towards combining the two images dimension-by-dimension whereas the agnostic sender does not (though we note the agnostic architecture nests the informed one).

---

[1]For example, Jorge et al. (2016) explore agents playing a "Guess Who" game to learn about the emergence of question-asking and answering in language.

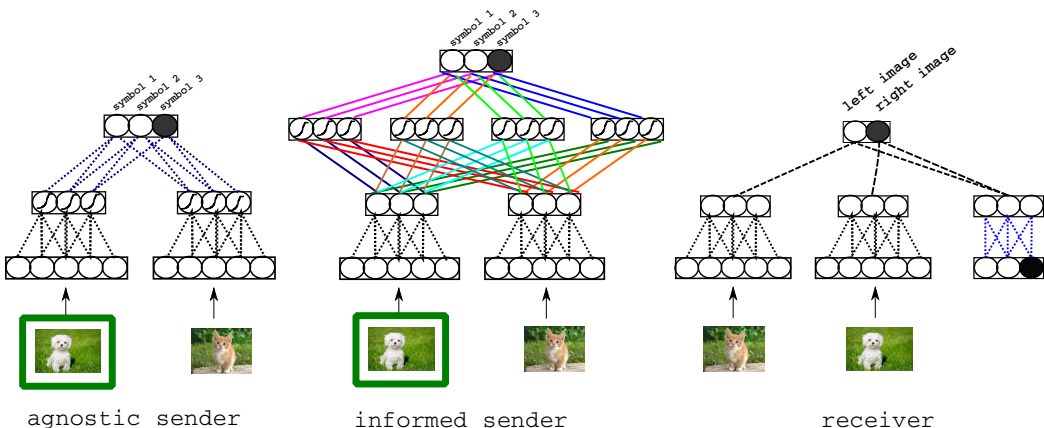

Figure 1: Architectures of agent players.

For both senders, motivated by the discrete nature of language, we enforce a strong communication bottleneck that discretizes the communication protocol. Activations on the top (vocabulary) layer are converted to a Gibbs distribution (with temperature parameter $\tau$), and then a single symbol $s$ is sampled from the resulting probability distribution.

The receiver takes as input the target and distractor image vectors in random order, as well as the symbol produced by the sender (as a one-hot vector over the vocabulary). It embeds the images and the symbol into its own "game-specific" space. It then computes dot products between the symbol and image embeddings. Ideally, dot similarity should be higher for the image that is better denoted by the symbol. The two dot products are converted to a Gibbs distribution (with temperature $\tau$) and the receiver "points" to an image by sampling from the resulting distribution.

**General Training Details** We set the following hyperparameters without tuning: embedding dimensionality: 50, number of filters applied to embeddings by informed sender: 20, temperature of Gibbs distributions: 10. We explore two vocabulary sizes: 10 and 100 symbols.

The sender and receiver parameters $\theta = \langle \theta_R, \theta_S \rangle$ are learned while playing the game. No weights are shared and the only supervision used is communication success, i.e., whether the receiver pointed at the right referent.

This setup is naturally modeled with Reinforcement Learning (Sutton & Barto, 1998). As outlined in Section 2, the sender follows policy $s(\theta_S(i_L, i_R, t)) \in V$ and the receiver policy $r(i_L, i_R, s(\theta_S(i_L, i_R, t))) \in \{L, R\}$. The loss function that the two agents must minimize is $-\mathbf{E}_{\tilde{r}}[R(\tilde{r})]$ where $R$ is the reward function returning 1 iff $r(i_L, i_R, s(\theta_S(i_L, i_R, t))) = t$. Parameters are updated through the Reinforce rule (Williams, 1992). We apply mini-batch updates, with a batch size of 32 and for a total of 50k iterations (games). At test time, we compile a set of 10k games using the same method as for the training games.

We now turn to our main questions. The first is whether the agents can learn to successfully coordinate in a reasonable amount of time. The second is whether the agents' language can be thought of as "natural language", i.e., symbols are assigned to meanings that make intuitive sense in terms of our conceptualization of the world.

## 4 LEARNING TO COMMUNICATE

Our first question is whether agents converge to successful communication at all. We see that they do: agents almost perfectly coordinate in the 1k rounds following the 10k training games for every architecture and parameter choice (Table 1).

We see, though, some differences between different sender architectures. Figure 2 (left) shows performance on a sample of the test set as a function of the first 5,000 rounds of training. The agents

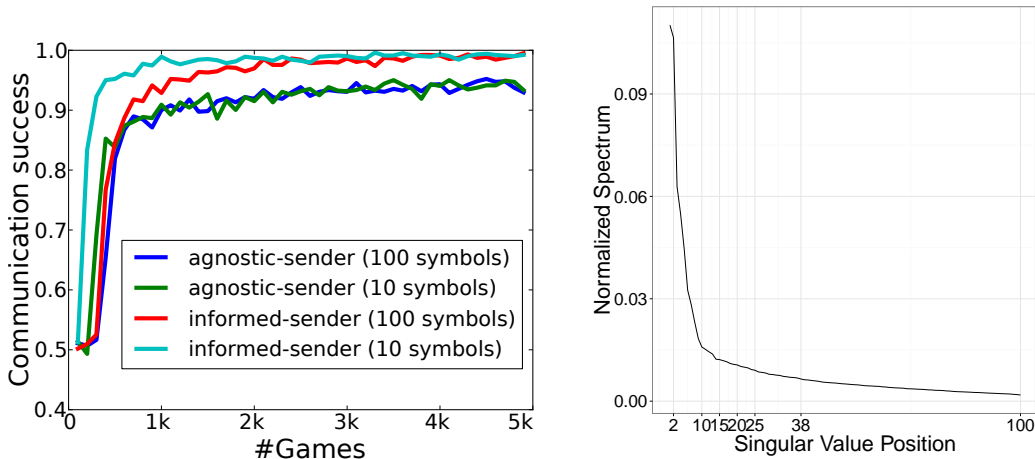

Figure 2: **Left:** Communication success as a function of training iterations, we see that informed senders converge faster than agnostic ones. **Right:** Spectrum of an example symbol usage matrix: the first few dimensions do capture only partial variance, suggesting that the usage of more symbols by the informed sender is not just due to synonymy.

| id | sender | vis rep | voc size | used symbols | comm success (%) | purity (%) | obs-chance purity (%) |
|---|---|---|---|---|---|---|---|
| 1 | informed | sm | 100 | 58 | 100 | 46 | 27 |
| 2 | informed | fc | 100 | 38 | 100 | 41 | 23 |
| 3 | informed | sm | 10 | 10 | 100 | 35 | 18 |
| 4 | informed | fc | 10 | 10 | 100 | 32 | 17 |
| 5 | agnostic | sm | 100 | 2 | 99 | 21 | 15 |
| 6 | agnostic | fc | 10 | 2 | 99 | 21 | 15 |
| 7 | agnostic | sm | 10 | 2 | 99 | 20 | 15 |
| 8 | agnostic | fc | 100 | 2 | 99 | 19 | 15 |

Table 1: Playing the referential game: test results after 50K training games. *Used symbols* column reports number of distinct vocabulary symbols that were produced at least once in the test phase. See text for explanation of *comm success* and *purity*. All purity values are highly significant ($p < 0.001$) compared to simulated chance symbol assignment when matching observed symbol usage. The *obs-chance purity* column reports the difference between observed and expected purity under chance.

converge to coordination quite fast, but the informed sender reaches higher levels more quickly than the agnostic one.

The informed sender makes use of more symbols from the available vocabulary, while the agnostic sender constantly uses a compact 2-symbol vocabulary. This suggests that the informed sender is using more varied and word-like symbols (recall that the images depict 463 distinct objects, so we would expect a natural-language-endowed sender to use a wider array of symbols to discriminate among them). However, it could also be the case that the informed sender vocabulary simply contains higher redundancy/synonymy. To check this, we construct a (sampled) matrix where rows are game image pairs, columns are symbols, and entries represent how often that symbol is used for that pair. We then decompose the matrix through SVD. If the sender is indeed just using a strategy with few effective symbols but high synonymy, then we should expect a 1- or 2-dimensional decomposition. Figure 2 (right) plots the normalized spectrum of this matrix. While there is some redundancy in the matrix (thus potentially implying there is synonymy in the usage), the language still requires multiple dimensions to summarize (cross-validated SVD suggests 50 dimensions).

We now turn to investigating the semantic properties of the emergent communication protocol. Recall that the vocabulary that agents use is arbitrary and has no initial meaning. One way to understand its emerging semantics is by looking at the relationship between symbols and the sets of images they refer to.

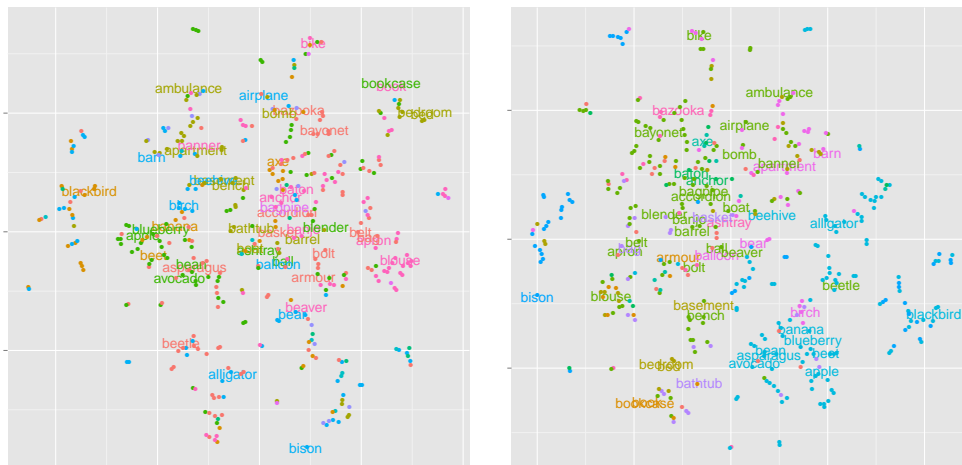

Figure 3: t-SNE plots of object fc vectors color-coded by majority symbols assigned to them by informed sender. Object class names shown for a random subset. **Left:** configuration of 4th row of Table 1. **Right:** 2nd row of Table 2.

The objects in our images were categorized into 20 broader categories (such as *weapon* and *mammal*) by McRae et al. (2005). If the agents converged to higher level semantic meanings for the symbols, we would expect that objects belonging to the same category would activate the same symbols, e.g., that, say, when the target images depict bayonets and guns, the sender would use the same symbol to refer to them, whereas cows and guns should not share a symbol.

To quantify this, we form clusters by grouping objects by the symbols that are most often activated when target images contain them. We then assess the quality of the resulting clusters by measuring their *purity* with respect to the McRae categories. Purity (Zhao & Karypis, 2003) is a standard measure of cluster "quality". The purity of a clustering solution is the proportion of category labels in the clusters that agree with the respective cluster majority category. This number reaches 100% for perfect clustering and we always compare the observed purity to the score that would be obtained from a random permutation of symbol assignments to objects. Table 1 shows that purity, while far from perfect, is significantly above chance in all cases. We confirm moreover that the informed sender is producing symbols that are more semantically natural than those of the agnostic one.

Still, surprisingly, purity is significantly above chance even when the latter is only using two symbols. From our qualitative evaluations, in this case the agents converge to a (noisy) characterization of objects as "living-vs-non-living" which, intriguingly, has been recognized as the most basic one in the human semantic system (Caramazza & Shelton, 1998).

Rather than using hard clusters, we can also ask whether symbol usage reflects the semantics of the visual space. To do so we construct vector representations for each object (defined by its ImageNet label) by averaging the CNN fc representations of all category images in our data-set (see Section 3 above). Note that the fc layer, being near the top of a deep CNN, is expected to capture high-level visual properties of objects (Zeiler & Fergus, 2014). Moreover, since we average across many specific images, our vectors should capture rather general, high-level properties of objects.

We map these average object vectors to 2 dimensions via t-SNE mapping (Van der Maaten & Hinton, 2008) and we color-code them by the majority symbol the sender used for images containing the corresponding object. Figure 3 (left) shows the results for the current experiment. We see that objects that are close in CNN space (thus, presumably, visually similar) are associated to the same symbol (same color). However, there still appears to be quite a bit of variation.

## 4.1 OBJECT-LEVEL REFERENCE

We established that our agents can solve the coordination problem, and we have at least tentative evidence that they do so by developing symbol meanings that align with our semantic intuition. We

| id | sender | vis rep | voc size | used symbols | comm success(%) | purity (%) | obs-chance purity (%) |
|----|--------|---------|----------|--------------|-----------------|------------|------------------------|
| 1  | informed | fc | 100 | 43 | 100 | 45 | 21 |
| 2  | informed | fc | 10  | 10 | 100 | 37 | 19 |
| 3  | agnostic | fc | 100 | 2  | 92  | 23 | 7 |
| 4  | agnostic | fc | 10  | 3  | 98  | 28 | 12 |

Table 2: Playing the referential game with image-level targets: test results after 50K training plays. Columns as in Table 1. All purity values significant at $p < 0.001$.

turn now to a simple way to tweak the game setup in order to encourage the agents to further pursue high-level semantics.

The strategy is to remove some aspects of "common knowledge" from the game. Common knowledge, in game-theoretic parlance, are facts that everyone knows, everyone knows that everyone knows, and so on (Brandenburger et al., 2014). Coordination can only occur if the basis of the coordination is common knowledge (Rubinstein, 1989), therefore if we remove some facts from common knowledge, we will preclude our agents from coordinating on them. In our case, we want to remove facts pertaining to the details of the input images, thus forcing the agents to coordinate on more abstract properties. We can remove all low-level common knowledge by letting the agents play only using class-level properties of the objects. We achieve this by modifying the game to show the agents different pairs of images but maintaining the ImageNet class of both the target and distractor (e.g., if the target is *dog*, the sender is shown a picture of a Chihuahua and the receiver that of a Boston Terrier).

Table 2 reports results for various configurations. We see that the agents are still able to coordinate. Moreover, we observe a small increase in symbol usage purity, as expected since agents can now only coordinate on general properties of object classes, rather than on the specific properties of each image. This effect is clearer in Figure 3 (right), when we repeat t-SNE based visualization of the relationship that emerges between visual embeddings and the words used to refer to them in this new experiment.

## 5 GROUNDING AGENTS' COMMUNICATION IN HUMAN LANGUAGE

The results in Section 4 show communication robustly arising in our game, and that we can change the environment to nudge agents to develop symbol meanings which are more closely related to the visual or class-based semantics of the images. Still, we would like agents to converge on a language fully understandable by humans, as our ultimate goal is to develop conversational machines. To do this, we will need to ground the communication.

Taking inspiration from AlphaGo (Silver et al., 2016), an AI that reached the Go master level by combining interactive learning in games of self-play with passive supervised learning from a large set of human games, we combine the usual referential game, in which agents interactively develop their communication protocol, with a supervised image labeling task, where the sender must learn to assign objects their conventional names. This way, the sender will naturally be encouraged to use such names with their conventional meaning to discriminate target images when playing the game, making communication more transparent to humans.

In this experiment, the sender switches, equiprobably, between game playing and a supervised image classification task using ImageNet classes. Note that the supervised objective does not aim at improving agents' coordination performance. Instead, supervision provides them with basic grounding in natural language (in the form of image-label associations), while concurrent interactive game playing should teach them how to effectively use this grounding to communicate.

We use the informed sender, fc image representations and a vocabulary size of 100. Supervised training is based on 100 labels that are a subset of the object names in our data-set (see Section 3 above). When predicting object names, the sender uses the usual game-embedding layer coupled with a softmax layer of dimensionality 100 corresponding to the object names. Importantly, the game-embedding layers used in object classification and the reference game are shared. Conse-

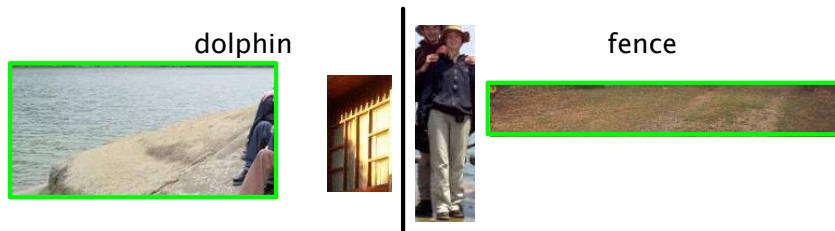

Figure 4: Example pairs from the ReferItGame set, with word produced by sender. Target images framed in green.

quently, we hope that, when playing, the sender will produce symbols aligned with object names acquired in the supervised phase.

The supervised objective has no negative effect on communication success: the agents are still able to reach full coordination after 10k training trials (corresponding to 5k trials of reference game playing). The sender uses many more symbols after training than in any previous experiment (88) and symbol purity dramatically increases to 70% (the obs-chance purity difference also increases to 37%).

Even more importantly, many symbols have now become directly interpretable, thanks to their direct correspondence to labels. Considering the 632 image pairs where the target gold standard label corresponds to one of the labels that were used in the supervised phase, in 47% of these cases the sender produced exactly the symbol corresponding to the correct supervised label for the target image (chance: 1%).

For image pairs where the target image belongs to one of the directly supervised categories, it is not surprising that the sender adopted the "conventional" supervised label to signal the target . However, a very interesting effect of supervision is that it *improves the interpretability of the code even when agents must communicate about images that do not contain objects in the supervised category set*. This emerged in a follow-up experiment in which, during training, the sender was again exposed (with equal probability) to the same supervised classification task as above, but now the agents played the referential game on a different dataset of images derived from ReferItGame (Kazemzadeh et al., 2014). In its general format, the ReferItGame contains annotations of bounding boxes in real images with referring expressions produced by humans when playing the game. For our purposes, we constructed 10k pairs by randomly sampling two bounding boxes, to act as target and distractor. Again, the agents converged to perfect communication after 15k trials, and this time used all 100 available symbols in some trial.

We then asked whether this language was human-interpretable. For each symbol used by the trained sender, we randomly extracted 3 image pairs in which the sender picked that symbol and the receiver pointed at the right target (for two symbols, only 2 pairs matched these criteria, leading to a set of 298 image pairs). We annotated each pair with the word corresponding to the symbol in the supervised set. Out of the 298 pairs, only 25 (8%) included one of the 100 words among the corresponding referring expressions in ReferItGame. So, in the large majority of cases, the sender had been faced with a pair not (saliently) containing the categories used in the supervised phase of its training, and it had to produce a word that could, at best, only indirectly refer to what is depicted in the target image. We then tested whether this code would be understandable by humans. In essence, it is as if we replaced the trained agent receiver with a human.

We prepared a crowdsourced survey using the CrowdFlower platform. For each pair, human participants were shown the two images and the sender-emitted word (that is, the ImageNet label associated to the symbol produced by the sender; see examples in Figure 4). The participants were asked to pick the picture that they thought was most related to the word. We collected 10 ratings for each pair.

We found that in 68% of the cases the subjects were able to guess the right image. A logistic regression predicting subject image choice from ground-truth target images, with subjects and words as random effects, confirmed the highly significant correlation between the true and guessed images

($z = 16.75$, $p < 0.0001$). Thus, while far from perfect, we find that supervised learning on a separate data set does provide some grounding for communication with humans, that generalizes beyond the conventional word denotations learned in the supervised phase.

Looking at the results qualitatively, we found that very often sender-subject communication succeeded when the sender established a sort of "metonymic" link between the words in its possession and the contents of an image. Figure 4 shows an example where the sender produced *dolphin* to refer to a picture showing a stretch of sea, and *fence* for a patch of land. Similar semantic shifts are a core characteristic of natural language (e.g., Pustejovsky, 1995), and thus subjects were, in many cases, able to successfully play the referential game with our sender (10/10 subjects guessed the dolphin target, and 8/10 the fence). This is very encouraging. Although the language developed in referential games will be initially very limited, if both agents and humans possess the sort of flexibility displayed in this last experiment, the noisy but shared common ground might suffice to establish basic communication.

## 6 DISCUSSION

Our results confirmed that fairly simple neural-network agents can learn to coordinate in a referential game in which they need to communicate about a large number of real pictures. They also suggest that the meanings agents come to assign to symbols in this setup capture general conceptual properties of the objects depicted in the image, rather than low-level visual properties. We also showed a path to grounding the communication in natural language by mixing the game with a supervised task.

In future work, encouraged by our preliminary experiments with object naming, we want to study how to ensure that the emergent communication stays close to human natural language. Predictive learning should be retained as an important building block of intelligent agents, focusing on teaching them structural properties of language (e.g., lexical choice, syntax or style). However, it is also important to learn the function-driven facets of language, such as how to hold a conversation, and interactive games are a potentially fruitful method to achieve this goal.

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
