# Peer review of "Multi-Agent Cooperation and the Emergence of (Natural) Language"

_ICLR 2017 — accepted_

[Official Review · AnonReviewer3 · rating 7 · confidence 3 · 16 Dec 2016]
soundness 4 · clarity 3

In this paper, a referential game is proposed between two agents. Both agents observe two images. The first agent, called the sender, receive a binary target variable (t) and must send a symbol (message) to the second agent, called the receiver, such that this agent can recover the target. The agents both get a reward, if the receiver agent can predict the target. The paper proposes to parametrize the agents as neural networks - with pretrained representations of the images as feature vectors - and train them using REINFORCE. In this setting, it is shown that the agents converge to  optimal policies and that their learned communications (e.g. the symbolic code transmitted from the sender to the receiver) have some meaningful concepts. In addition to this, the paper presents experiments on a variant of the game grounded on different image classes. In this setting, the agents appear to learn even more meaningful concepts. Finally, multi-game setup is proposed, where the sender agent is alternating between playing the game before and playing a supervised learning task (classifying images). Not surprisingly, when anchored to the supervised learning task, the symbolic communications have even more meaningful concepts.

Learning shared representations for communication in a multi-agent setup is an interesting research direction to explore. This is a much harder task compared to standard supervised learning or single-agent reinforcement learning tasks, which justifies starting with a relatively simple task. To the best of my knowledge, the approach of first learning communication between two agents and then grounding this communication in human language is novel. As the authors remark, this may be an alternative paradigm to standard sequence-to-sequence models which tend to focus on statistical properties of language rather than their functional aspects. I believe the contributions of the proposed task and framework, and the analysis and visualization of what the communicated tokens represent is a useful stepping stone for future work. For this reason, I think the paper should be accepted.



Other comments:
- How is the target (t) incorporated into the sender networks? Please clarify this.
- Table 1 and Table 2 use percentage (%) values differently. In the first, percentages seem to be written in the interval [0, 100], and in the second in the interval [0, 1]. Please correct this. Perhaps related to this, in Table 1, the column "obs-chance purity" seems to have extremely small values. I assume this was mistake?
- "assest" -> "assess"
- "usufal" -> "usual"

[Official Review · AnonReviewer2 · rating 7 · confidence 3 · 21 Dec 2016 (modified: 25 Jan 2017)]
appropriateness 2

To train natural language systems by putting multiple agents within an interactive referential communication game is very nice. As the authors mention, there has been some (although seemingly not much) previous work on using multi-agent games to teach communication, and it certainly seems like a direction worth pursuing. Moreover, the approach of switching between these games and some supervised learning, as in the experiment described in Section 5 and suggested in Section 6, seems particularly fruitful. 

Note: For “clarity”, I believe some of the network connections in Fig 1 have been omitted. However, given the rather highly-customized architecture and the slightly hard-to-follow description in Section 3, the shorthand diagram only adds to the confusion. The diagram probably needs to be fine-tuned, but at the very least (especially if I am misunderstanding it!), a caption must [still] be added to help the reader interpret the figure. 

Overall, the framework (Section 2) is great and seems quite effective/useful in various ways, the results are reasonable, and I expect there will be some interesting future variations on this work as well.

Caveat: While I am quite confident I understood the paper (as per confidence score below), I do not feel I am sufficiently familiar with the most closely related literature to accurately assess the place of this work within that context.

[Official Review · AnonReviewer1 · rating 7 · confidence 3 · 23 Dec 2016]
**Interesting idea, but could have been solved using a transfer learning approach**
originality 4 · clarity 5 · impact 4

Thank you for an interesting read.

Pros
- This paper tackles a very crucial problem of understanding communications between 2 agents. As more and more applications of reinforcement learning are being explored, this approach brings us back to a basic question. Is the problem solving approach of machines similar to that of humans.

- The task is simple enough to make the post learning analysis intuitive.

- It was interesting to see how informed agents made use of multiple symbols to transmit the message, where as agnostic agents relied only on 2 symbols. 

Cons
- The task effectively boils down to image classification, if the 2 images sent are from different categories. The symbols used are effectively the image class which the second agent learns to assign to either of the images. By all means, this approach boils down to a transfer learning problem which could probably be trained much faster than a reinforcement learning algorithm.

[Public Comment · Katja Filippova · 07 Feb 2017]
**Interesting paper, some important references seem to be missing**

The paper is interesting and well-written. The topic of language evolution has fascinated researchers from so many different fields (all kinds of linguistics, anthropology, psychology, sociology, but even mathematics and computer science), and now it looks like there is a new surge of interest -- even a long-time sceptic Noam Chomsky recently published a book. 

It is a pity the paper is rather weak on references and related work. Not on language evolution in general, but on modeling language evolution as a multi-agent cooperation. For example, the submission does not even mention the Lewis Signaling Game which is basically what the paper proposes (

[Public Comment · Jonathan Bonnet · 08 Feb 2017]
**interesting paper, should enhance section about cooperation**

Very interesting paper. Maybe you could read paper by Gleizes about cooperative agent, self-organization and resolution through emergence … This paper could interest you : Self-adaptive complex systems, Marie-Pierre Gleizes, 2011, EUMAS.

[Final Decision · Program Chairs · 06 Feb 2017]
**ICLR committee final decision**

The authors present some initial findings on language emergence using multi-agent, referential games. The learning alternates between REINFORCE and supervised classification, which grounds the language. Pro- this is a relevant, novel paper. Con - experiments are somewhat simple/limited.